# Supplementation of Plants with Immunomodulatory Properties during Pregnancy and Lactation—Maternal and Offspring Health Effects

**DOI:** 10.3390/nu11081958

**Published:** 2019-08-20

**Authors:** Aneta Lewicka, Łukasz Szymański, Kamila Rusiecka, Anna Kucza, Anna Jakubczyk, Robert Zdanowski, Sławomir Lewicki

**Affiliations:** 1Laboratory of Epidemiology, Military Institute of Hygiene and Epidemiology, Kozielska 4, 01-163 Warsaw, Poland; 2Department of Microwave Safety, Military Institute of Hygiene and Epidemiology, Kozielska 4, 01-163 Warsaw, Poland; 3Department of Regenerative Medicine and Cell Biology, Military Institute of Hygiene and Epidemiology, Kozielska 4, 01-163 Warsaw, Poland; 4Department of Biochemistry and Food Chemistry, University of Life Sciences, Skromna 8, 20-704 Lublin, Poland

**Keywords:** immunomodulators, maternal nutrition, pregnancy, *Rhodiola*, *Echinacea*, *Panax*, *Camellia*

## Abstract

A pregnant woman’s diet consists of many products, such as fruits, vegetables, cocoa, tea, chocolate, coffee, herbal and fruit teas, and various commercially available dietary supplements, which contain a high number of biological active plant-derived compounds. Generally, these compounds play beneficial roles in women’s health and the development of fetus health. There are, however, some authors who report that consuming excessive amounts of plants that contain high concentrations of polyphenols may negatively affect the development of the fetus and the offspring’s health. Important and problematic issues during pregnancy and lactation are bacterial infections treatment. In the treatment are proposals to use plant immunomodulators, which are generally considered safe for women and their offspring. Additional consumption of biologically active compounds from plants, however, may increase the risk of occurrences to irreversible changes in the offspring’s health. Therefore, it is necessary to carry out safety tests for immunomodulators before introducing them into a maternal diet. Here, we present data from animal experiments for the four most-studied plants immunomodulators genus: *Rhodiola*, *Echinacea*, *Panax*, and *Camellia*, which were used in maternal nutrition.

## 1. Introduction

The antibiotic treatments of bacterial infections during pregnancy and lactation are very problematic because of their negative effects on embryo and newborn development. The food and drug administration agency divided all known antibiotics into five groups (A, B, C, D, X) depending on the potential risk of harmful effects on the offspring. In general, only group A antibiotics are considered to be safe when used during pregnancy and lactation [1]. Prenatal exposure to the antibiotics from the remaining groups may modify the immunological system of the offspring. The offspring of mothers treated during pregnancy with the group B antibiotics, such as penicillin and cephalosporin, showed, after pathogen stimulation, a decrease of the specific immune response and increase of the nonspecific immune response [2]. Also, prenatal treatment with antibiotics increased the risk of childhood asthma and obesity [3]. The alternative therapy, which could potentially support or even replace antibiotic treatment, would be the use of plant-derived immunomodulators. Nowadays, there are more than 300 plants which exhibit beneficial properties for humans. Plant-derived compounds which affect the immune system belong to chemical groups of compounds such as: alkaloids (i.e., leonurine [4], piperine [5], and sophocarpine [6]), terpenoids (i.e., ginsan [7] or oleanolic acid [8]), polysaccharides (from licorice *Glycyrrhiza uralensis Fisch* [9], or from *Lentinus edodes* [10]), lactones (from *Datura quercifolia* [11] or sesquiterpene lactones from *Loranthus parasiticus*), and essential oils (Z-ligustilide [12] or Tetramethylpyra-zine [13]). However, the most well-known and numerous group of immune-affecting compounds belongs to flavonoids, which are divided into flavones, flavanones, flavanols, flavonols, catechins, isoflavones, and anthocyanins [14]. Some of the flavonoids, such as genistein or quercetin, are commonly-used supplements in human nutrition [15,16]. Plant-derived immunomodulators affect the immune system in various ways: [1] directly by affecting the immune cell functions in both adaptive and innate immunity [17] and by direct anti-microbial, anti-viral, or anti-fungal properties [18] or [2] indirectly by modulation of non-immune cell function [19], reduction of inflammation [20], the influence of cytokines secretion [21], or modulation of angiogenesis [22]. Most importantly, plant-derived immunomodulators are generally considered as safe for humans and therefore are a good alternative for the use of most synthetic immunomodulators, which cause various side effects. This statement is mostly true for people in a normal state of life, however, for pregnant women and their offspring, this can become dangerous. Several researchers had shown that theobromine and cocoa catechins administered to pregnant mice affected embryonic angiogenesis and morphology and function of some of their progeny organs, among them the lymphoid system and kidneys [23,24,25,26,27]. Therefore, before the recommendation of supplementation of plant-derived immunomodulators in the maternal diet, experiments on animals should be performed. Here, we presented data from animal experiments for the four most-studied plants’ immunomodulators genus, *Rhodiola*, *Echinacea*, *Panax*, and *Camellia*, which were used in maternal nutrition. The summarized effects of Rhodiola, *Echinacea*, Ginseng, and Camellia, or its extracts supplemented in pregnancy and/or lactation on mothers and offspring health, are presented in Table 1.

## 2. Rhodiola

### 2.1. Characteristic and Immunomodulatory Properties

Genus *Rhodiola* (family *Crassulaceae*) consists of 200 species, out of which over 20 species have medicinal properties. *Rhodiola* extracts contain phenylethanoid salidroside and tyrosol, phenolic acids (i.e., chlorogenic, ferulic, ellagic, and p-coumaric), and flavonoids (i.e., fisetin, naringenin, kaempferol, epicatechin, luteolin, quercetin, epigallocatechin, and (+)-catechin) [63,64,65]. They have been used in traditional Asian and European medicine such as tonics, adaptogens, antidepressants, and anti-inflammatory compounds [66]. The extracts from these plants also have anticancer, antibacterial, and immunomodulatory properties [67,68,69,70]. *Rhodiola* extracts, given for seven days to mice infected with *Pseudomonas aeruginosa*, decrease the infection. These mice have a higher number and higher metabolic activity of blood leukocytes [71]. *Rhodiola* extracts also have a direct antiviral and antibacterial effect against hepatitis C virus (HCV) and *Mycobacterium tuberculosis* [72,73]. At present, there are several clinical therapies available, which use *Rhodiola* plants to stimulate the immune system [74]. One of the well-characterized plant species from the *Rhodiola* genus which was supplemented in pregnancy and lactation is *Rhodiola kirilowii*, which was supplemented as lyophilized water extract (RKW) or 50% hydro-alcoholic extract (RKW-A) in the pregnancy and lactation period in mice (about 42 days of supplementation).

### 2.2. Effects on Mothers

Supplementation of mice during pregnancy and lactation with water extract (RKW) or 50% hydro-alcoholic extract (RKW-A) of *Rhodiola kirilowii* (concentration 20 mg extract/kg body weight) had no effect on the average body weight as compared to the control group (receiving sterile water). There were no differences in the mean weight and mass index of selected organs: liver, kidneys, spleen, brain, and eyeballs, between the study groups. Macroscopic assessment of organs also did not reveal any changes in their structure. The morphological elements of blood: WBC (number of white blood cells/mm^3^), RBC (number of red blood cells/mm^3^), HGB (hemoglobin, g/dL), HCT (hematocrit, %), MCH (mean corpuscular hemoglobin, pg), MCHC (mean hemoglobin concentration, g/dL), RDW (red blood cell distribution width, %), PLT (platelet count /mm^3^), and MPV (mean platelet volume, fL), did not differ between the control group and the groups receiving extracts from *Rhodiola kirilowii*. There were no differences in the population of cells belonging to the adaptive immunity system, T cells (CD3+, CD4+, CD8+) and B cells (CD19+), in the blood. However, some small changes in the composition and functioning of innate immunity cells were noticed. Mice fed with RKW extracts showed a lower percentage of cells with a respiratory burst in granulocytes (PhagoBurst test) [28]. In contrast, supplementation with RKW-A extract caused an increase in the percentage of granulocytes and monocytes in the blood with the respiratory burst. Other components of non-specific immunity have not changed (including the number and percentage of NK (natural killer) cells and the percentage of phagocytic cells). There were no changes in the concentration of selected cytokines in the serum: Th1 (IL-2, TNF-α, INF-γ), Th2 (IL-4, IL-6, IL-10), and Th17 (IL-17a) [30].

Despite the lack of differences in the weight and mass index of the spleen in mothers fed with *Rhodiola kirilowii* extracts, significant changes in spleen morphology and structure were observed. Spleens from mothers fed during pregnancy and the lactation water or hydroalcoholic extracts from *Rhodiola kirilowii* contained a significantly lower number of cells per gram of organ [29]. In the cytometric study, a significant reduction in the percentage of lymphocytes was associated with an increase in the percentage of monocytes and granulocytes in the spleen only in the group receiving a 50% hydro-alcoholic extract of *Rhodiola kirilowii*. However, no significant changes in the percentage of the innate (CD335+) and adaptive (CD3+, CD4+, CD8+, CD19+) cell populations between the study groups were observed. Analysis of the total number of cell populations per gram of organ showed no difference in the number of CD3+ and CD8+ cells between groups. However, we found that both extracts reduced the number of CD4+ and CD19+ cells. In addition, water extract significantly reduced the total number of NK cells (CD335+) [31]. The observed changes in the number of cells in the spleen suggest that treatment caused a certain degree of impairment of the adaptive immune response. Therefore, we decided to examine the spleen cells’ (mainly T and B cells) proliferation response to mitogens stimulations (ex vivo studies). We found that there were no significant differences in the proliferative activity of splenocytes stimulated with lipopolysaccharide (LPS), phytohemagglutinin (PHA), or concanavalin A (ConA). This means that the functionality of these cells, despite the lower number in the spleen, has been preserved. However, the observed, unfavorable reduction in the number of splenocytes is a disturbing sign, which should be taken into account in the long-term administration of extracts from *Rhodiola kirilowii* to pregnant women [29,30].

The RKW-A group had a higher concentration of VEGF (vascular endothelial growth factor) and bFGF (basic fibroblast growth factor) than the control group. These trends were associated with decreased proliferation of endothelial cells (HECa10) after the supplementation of medium with the serum isolated from mothers fed during pregnancy and lactation with (RKW-A). However, the RKW group showed increased migration of HECa10 cells [30]. The obtained results led to the conclusion that supplementation of hydro-alcoholic extracts of *Rhodiola kirilowii* may cause modulation of angiogenesis processes in developing fetuses. A similar effect was observed in previous studies in which RKW-A extract inhibited tumor angiogenesis and the RKW had no effect [63].

In conclusion, slight changes in adaptive immunity (blood) and in the number of splenocytes with a lack of side effects on the range of morphological blood parameters of mother mice should be considered as a positive effect of supplementation, with such a long period of use. Usually, plant-derived immunomodulatory drugs are used for no more than 2–3 weeks. This is due to the fact that prolonged administration of immunostimulants may cause lack of stimulation, and in extreme situations, deregulations of the immune system, as with what was seen in aquaculture [75]. Authors suggested that for successful use of immunostimulators, not only appropriate timing of administration, but also dosage and period of administration are necessary.

### 2.3. Effects on Offspring

The animal mass analysis did not show a significant difference between the offspring from the control group and the group supplemented with water extract of *Rhodiola kirilowii*. The mean weight of offspring whose mothers were fed with 50% hydro-alcoholic extract was about 7% lower than in the control group (*p* < 0.01). There were no significant differences in peripheral blood morphometric parameters (WBC, RBC, HCT, MCH, MCHC, RDW, RDW-a, PLT, and MPV) between the control group and the groups receiving *Rhodiola kirilowii* extracts. There was a slight, but significant, increase in hemoglobin concentration (about 0.6 mg/dL) in the group supplemented with *Rhodiola kirilowii* water extract [32].

The analysis of selected components of the adaptive and innate immune system in the blood and spleens of offspring, whose mothers were fed during pregnancy and lactation extracts from *Rhodiola kirilowii*, showed significant differences. A significant decrease in the mean percentage of lymphocytes (approximately 6%, *p* < 0.05) in peripheral blood and an increase in the mean percentage of granulocytes (approximately 20%, *p* < 0.05) were observed in the RKW group (a similar trend was observed in the RKW-A group, however, it was not statistically significant). The analysis of the lymphocyte population showed a significant decrease in the percentage of CD3+ cells (RKW-14%, *p* = 0.0434, RKW-A-10%, *p* = 0.0337) and CD4+ (15%, *p* = 0.0184 and 13%, *p* = 0.0116, respectively). In addition, a lower percentage of CD8+ cells and a higher percentage of NK cells (CD335+) were observed in the RKW group. There were no changes in the percentage of B cells (CD19+) and the percentage of regulatory T cells (CD4+, CD25+, FoxP3+) and CD4+, CD25+ cells between the examined groups [32].

A significant decrease in the percentage of CD4+ in the peripheral blood may indicate a certain impairment of adaptive immunity, which may lead to the disorders in the stimulation of CD19+ cells and the production of specific antibodies. In part, these results were confirmed in the SRBC (sheep red blood cells) test (immunization of mice with sheep red blood cells), in which much lower levels of anti-SRBC antibodies were observed in the serum of mice whose mothers were fed during pregnancy and lactation with RKW-A [33]. The explanation of the lower immune response after RKW-A supplementation may also be associated with an increased concentration of IL-10 in the serum. This cytokine has anti-inflammatory activity and inhibits the synthesis of other pro-inflammatory cytokines that are necessary to activate the immune system in response to antigens. No such changes were observed for water extract (RKW) [33].

Both extracts of *Rhodiola kirilowii* consumed during pregnancy and lactation by mothers stimulated the phagocytosis process in their offspring. There was an increase in the phagocytosis of opsonized *E. coli* bacteria and the intensity of the oxygen burst after stimulation with *E. coli* and zymosan [32]. Interestingly, in the RKW group, there was a significant reduction in serum IL-17a concentration [33]. IL-17a is a cytokine responsible inter alia for supporting phagocytosis of extracellular bacteria. It seems that the observed situation may be related to the activation of granulocytes by polyphenolic compounds present in the extracts, which in turn leads to a reduction of IL-17a secretion (immune system is not “over-activated”). The lack of additional stimulation by IL-17a could also lead to a decrease in the population of phagocytic monocytes and oxygen burst after zymosan stimulation in the RKW group [33].

Spleens of mice fed with *Rhodiola kirilowii* extracts did not differ in mass and mass index (spleen weight/mouse weight) and cellularity compared to the spleens obtained from the control group. There were no significant differences in the morphology of the spleens. In the cytometric study of spleen cells, no significant differences in the percentage of CD3+, CD4+, CD8+, CD19+, and CD335+ cells were noticed [32]. Histological analysis showed no significant difference in the number and location of CD4+ cells, but a significant difference in tissue localization and the number of CD8+ cells. These cells were found not only in usual locations within the spleen but also in the perivascular lymphatic sheath (PALS), the follicular zone of B cells, and in the red pulp. In addition, there was a much higher number of CD8+ cells in the central part of the spleen [34]. This may indicate increased mobility of these cells and thus increased ability to respond to antiviral responses. Supplementation of mothers with extracts from *Rhodiola kirilowii* during pregnancy and lactation also influenced cell proliferation in response to mitogen supplementation: lipopolysaccharide (LPS), phytohaemagglutinin (PHA), and concanavalin A (ConA). A decrease in cell proliferation rate was observed in the RKW-A group after the supplementation of LPS and PHA when compared to the control group. There was a reduction of proliferation rate in the RKW-A group after stimulation with ConA when compared to the RKW group [32]. The presented data indicated attenuation of the ability of adaptive immunity in the hydro-alcoholic *Rhodiola kirilowii* group.

Changes in the percentage of blood cells associated with adaptive immune response (mainly CD3+) suggested a possible negative effect on the maturation of these cells in the thymus in the offspring. The mean thymus weight and the thymic index did not differ between the groups. There were no significant differences in the morphology of the thymus between groups. There were also no abnormalities in the structure of lobules, medulla, cortex ratio, and epithelial cell content. No thymic hypertrophy or atrophy was detected. The offspring of mice whose mothers were fed during pregnancy and lactation RKW and RKW-A extracts had a significantly lower number of apoptotic cells (marker M30) than the control group. However, there was no significant difference in the number of cells synthesizing IL-7 [35]. The obtained data led to the conclusion that extracts from *Rhodiola kirilowii* prolonged the function of thymus cells in the offspring, which may be important for overcoming the infection.

Moreover, Lewicki et al. (2017) [76] found that there were no significant differences in the mass index and macroscopic structure of the kidneys between the examined groups. A higher percentage of individuals with serum creatinine above 0.65 mg % was found in the RKW and RKW-A groups. In addition, the offspring of mothers from RKW-A group had slightly increased urea concentration and decreased cystatin C concentration. These data suggested some abnormalities in the kidneys. In the RKW-A group, a higher number of tufts per mm^2^ was observed and their smaller diameter may indicate their atrophy. These changes may have been caused directly by the active substances in the extracts from *Rhodiola kirilowii* present in the maternal sera, and/or indirectly by affecting the secretion of proangiogenic factors. The differences were observed in the concentration of polyphenols in mice-mothers supplemented with RKW or RKW-A (mainly catechin and salidroside), a lower concentration of VEGF in the sera of the offspring from the RKW-A group [30]. These data indicate that hydro-alcoholic extract should not be long-term supplemented to pregnant women.

## 3. Echinacea

### 3.1. Characteristic and Immunomodulatory Properties

*Echinacea* is a plant belonging to the species from the *Asteraceae* family [77]. There are three species of *Echinacea* with medicinal properties, including *Echinacea purpurea*, *Echinacea pallida*, and *Echinacea angustifolia* [78]. *Echinacea* extracts contained alkamides, ketoalkenes, caffeic acid derivatives, polysaccharides, glycoproteins, and caftaric acid, which are responsible for the medical activity of the plant.

*Echinacea*-based drugs contribute to a shortening of the various types of infections and colds [79,80]. This is due to the active ingredients such as polysaccharides, alkaloids, coffee acid derivatives [81], and proteoglycans that have immunomodulatory, antiviral, antioxidant, and anti-inflammatory properties [78]. Polysaccharides, especially arabinogalactans, activate macrophages and have a cytotoxic effect against cancer cells [82,83]. During this process, various biological products are formed [84], including nitric oxide, which has a defensive function in the immune system [85]. Bacterial infections contribute to the formation of the inflammatory mediators, which in turn leads to the increase of the nitric oxide levels [86]. This compound is very harmful to healthy cells [87,88]. Studies have shown that the alcohol extracts of the *Echinacea purpurea*, *Echinacea pallida*, and *Echinacea angustifolia* significantly reduce the production of nitric oxide [86]. The extract also increased the number of non-activated macrophage cells that are the first to defend the body against infections [86]. The active ingredients present in *Echinacea* also show antioxidant activity [89,90,91,92]. Such components include the alkamids that have an antioxidant effect on the coffee acid derivatives [92].

### 3.2. Effects on Mothers

Dabboul et al. (2016) conducted a study to check the reproductive and immune parameters of pregnant rabbits during *Echinacea pallida* diet supplementation [36]. The studies were performed on 100 pregnant rabbits at the age of 21 weeks and supplemented with 3 g/day of *Echinacea pallida* for 4 weeks. The tests were carried out on days 0, 14, and 28 of the experiment. The effect of this plant on blood morphology parameters of mothers: red blood cells (n/mm^3^), hemoglobin (g/dL), hematocrit (%), mean corpuscular volume (fL), mean corpuscular hemoglobin (pg), mean hemoglobin concentration (g/dL), red blood cell distribution width (%), platelet count (n/mm^3^), mean thrombocyte volume (PCT,%), mean platelet volume (fL), platelet distribution width (%), number of white blood cells (n/mm3), lymphocytes (%), monocytes (%), neutrophils (%), eosinophils (%), and basophils (%) has not been observed.

The concentration of total protein (g/dL), glutamic oxoloacetic transaminase (UI/L), blood urea nitrogen (mg/dL), albumin (g/dL), urea (mg/dL), and cholesterol (mg/dL) were also analyzed. The serum protein was determined using semi-automatic agarose gel electrophoresis. Blood and immune parameters were determined in three periods (day 0, day 14, and day 28) and were subjected to statistical analysis using the GLM thematic model. In the case of reproductive parameters, the analysis was carried out for the specific *Echinacea* test and the control sample using the Student’s *t*-test. The results of the study showed that *Echinacea* supplementation does not affect hematological and reproductive parameters. Also, in the case of immune parameters, there were no significant differences between the test and control groups.

In another study carried out by Maass et al. (2005), dried *Echinacea purpurea* was introduced in the diet of sows and its effect on the results of plasma enzymes, blood, lymphocyte proliferation, antibodies, and immunoglobulin content in colostrum were checked [37]. The experiments were carried out with the use of various concentrations of *Echinacea* in the diet during different periods of pregnancy and lactation. The period of *Echinacea* supplementation from days 85–110 of pregnancy and in the 4th, 6th, and 9th week of lactation was considered. The study was carried out on 36 sows from the 85th day of pregnancy to the 28th day of lactation. Animals were divided into three experimental groups by weight. The diet during pregnancy and lactation was supplemented by 0%, 1.2%, or 3.6% and 0%, 0.5%, or 1.5% *Echinacea*. Due to the lack of data for pigs, the dose of *Echinacea* was given based on human recommendations (16.5 mg of freshly squeezed juice/kg). Blood analysis was performed on day 85 of pregnancy as well as on day 1 and 28 of lactation. The colostrum was collected manually from 1 to 6 hours after delivery.

Body mass, body temperature, health status, crude protein (6.37 *Kjeldahl-N), immunoglobulins, hematological, and clinic–chemical parameters (alkaline phosphatase, alanine, and aspartate, and gamma-glutamyltransferase aminotransferase) were analyzed using acquired samples. No influence of *Echinacea* on the enzyme results (alkaline phosphatase (U/L), alanine aminotransferase (U/L), aspartate aminotransferase (U/L), gamma-glutamyl transferase (U/L)) was observed. The level of crude protein in colostrum was lower in the *Echinacea* supplemented group. There was no statistical difference in the levels of leukocytes (10^9^/L), erythrocytes (10^12^/L), lymphocytes (%), granulocytes, neutrophils (%), eosinophils (%), basophils (%), or monocytes (%) between the study (*Echinacea* supplemented) and control group. Also, similar results were obtained in a group of breeding pigs. The experiment included two phases of *Echinacea* supplementation (1–3 weeks and 7–9 weeks) and an intermediate phase without supplementation (4–6 weeks). The experiment aimed to investigate the immune effect and the production of antibodies during vaccination of pigs. A differential vaccine was given, which was administered at week 1 and week 5 of the experiment. IgG1 immunoglobulin in colostrum was determined using ELISA technique. From the obtained data, such as the hematological analysis and the varied number of blood cells, supplementation had no effect. The level of antibodies in the plasma showed a significant effect of *Echinacea* in relation to all antibodies, which caused a significantly higher immune response in the control group. The health of the animals was also good.

Chow et al. (2006) focused on studying the effects of *Echinacea* on immunity and spontaneous miscarriages. The study was conducted in mice in which *Echinacea* was fed from the beginning to the 10th, 11th, 12th, 13th, and 14th days of pregnancy [38]. The mother’s spleen and bone marrow were collected for examination. The effect of *Echinacea* on hematopoietic cells in the spleen and bone marrow was determined using the Student’s *t*-test. The significant differences were found in the results obtained from the immune system analysis in the third trimester in the spleen of pregnant mice. The parameters and number of spleen lymphocytes and nucleated erythroid cells were decreased in *Echinacea*-fed mice. The bone marrow parameters were not influenced by the *Echinacea* supplementation. The results also indicated that miscarriages are more likely to happen in the early stages of pregnancy (10–11 days) in the *Echinacea*-fed mice. Based on these results, it was decided not to suggest the consumption of *Echinacea* in the early stages of pregnancy in women. On the other hand, the results of clinical trials conducted by Heitman et al. (2016) did not show adverse effects of *Echinacea* supplementation of mothers during pregnancy [41].

### 3.3. Effects on Offspring

Despite the large number of papers which confirmed the immunomodulatory effect of *Echinacea* spp. and a lot of products on the market containing *Echinacea*, there is still a little evidence of offspring safety after uses of the herb in pregnancy and lactation. In 2007, Barcz et al. investigated the effect of alcoholic extracts of *Echinacea* purpurea given to pregnant mice on angiogenic activity and tissue VEGF and bFGF production of their fetuses. Eight pregnant females, from the 1st to the 18th day of pregnancy, were given a 0.06 g solution of *Echinacea* purpurea from different formulations (three mothers were given Esberitox, another three were Echinapur, while two mothers were fed Immunalforte). On day 18, the females were sacrificed, and the fetuses were used for the angiogenesis test. It was found that two *Echinacea* drugs lowered the number of embryos in litter and significantly diminished the vascular endothelial growth factor (VEGF) and the basic fibroblast growth factor (bFGF) content of embryos tissue [39]. The effect of mothers’ supplementation in pregnancy and lactation of *Echinacea pallida* on the health of their offspring was also studied by Kovitvadhi et al. (2016) [40]. The authors supplemented sows with *Echinacea* from days 85–110 of pregnancy and in the 4th, 6th, and 9th week of lactation. Offspring from those mothers exhibited an increased phagocytic in blood activity and a higher bacterial diversity compared to other groups. There was no statistically significant difference in animal growth, blood parameters, and humoral immune response against vaccination or against the rabbit hemorrhagic disease virus.

The data from the human studies revealed no effect of *Echinacea* treatment in pregnancy and lactation in mothers on the health of their offspring. The Gallo et al. (2000) study also showed no effect of consuming *Echinacea* during pregnancy on the increased risk of malformations [42]. Also, Perri et al. (2006), in a prospective cohort study, concluded that *Echinacea* is non-teratogenic when used during pregnancy and lactation. The conclusions were based on a lack of evidence that maternal *Echinacea* consumption varied affected major or minor birth defects, differences in pregnancy outcome, delivery method, maternal weight gain, gestational age, infant birth weight, or fetal distress. It should be noted that daily dosage was varied, however, *Echinacea* supplementation was usually used by mothers for up to 7 days [43]. In a cohort study, Heitmann et al. (2016) evaluated the impact of prenatal exposure to *Echinacea* and the consequences of its use on malformations and adverse pregnancy outcomes. Based on the analysis of questionnaires completed by pregnant women at 17 and 30 weeks of pregnancy, 6 months after birth, and information on pregnancy results from the Norwegian birth register, there was no increased risk of malformations or adverse delivery results, such as premature delivery [41].

## 4. Ginseng

### 4.1. Characteristic and Immunomodulatory Properties

Ginseng belongs to angiosperms, the *Araliaceae* family, *Panax L.* genus. There are thirteen ginseng species, the most popular of which are *Panax ginseng* C.A. Meyer, *Panax quinquefolium* L., *Panax japonicus* C.A. Meyer. For thousands of years, it has been used in East Asia as a medicinal plant due to the active substances found in the root. These are polysaccharides, flavonoids, fatty acids, peptides, and saponins [93].

The main active substances in the ginseng root are ginsenosides. These are chemical compounds that belong to saponins. Due to the structure, there are three main groups: the protopanaxadiol group, the oleanane group, and the protopanaxatriol group [94]. Ginsenosides have been shown to have anti-inflammatory properties, mainly by inhibiting the production of TNF-α in a mouse macrophage cell line RAW264.7, that was exposed to lipopolysaccharide stimulation [95]. Lee et al. (2005) showed that 20-*O*-beta-d-glucopyranosyl-20 (*S*)-protopanaxadiol inhibits TPA-induced expression of COX-2, which in turn may contribute to antitumor activity [96]. Also, Ginsenoside-Re can inhibit the interaction between LPS and TLR4 (toll-like receptor 4) [97]. Other important active substances of ginseng root are polyphenols and polysaccharides [98]. Byeon et al. (2012) have shown that ginseng derived polysaccharides can be used as an immunostimulant via TLR2 (toll-like receptor 2), which mediates the activation of macrophages [99]. Also, TLR2 mediated functional activation of macrophages can be boosted by wortmannin-targeted enzymes. Ginseng polysaccharides also have immunomodulatory, anticancer, and antidiabetic effects [100].

### 4.2. General Effect of Ginseng

For thousands of years, ginseng root has been used as an immuno-stimulating plant. With the development of science, further properties and mechanisms of ginseng’s actions have been discovered. The administration of the ginseng extract has a positive effect on the spinal cord injury and thus has a neuroprotective effect [101]. However, it has been proven that the use of ginseng extract during pregnancy may have a teratogenic effect. According to research by Khalid et al. (2008), a high dose of the extract may also lead to the development of bone defects [102].

The literature describes many mechanisms of action of ginseng on the immune system. Kim et al. (2009) found that ginsan, ginseng-derived polysaccharide, has immunomodulatory effects on dendritic cells. They showed that ginsan stimulates dendritic cells by inducing their maturation. Ginsan stimulates secretion of cytokines from dendritic cells, increases the proliferation of allogeneic CD4 + lymphocytes, and enhances the expression of CD86 on the surface of dendritic cells [103]. Shin et al. (2002) have shown that ginseng acidic polysaccharides enhance the phagocytic activity of macrophages, stimulates cytokine secretion, and leads to the increased CD14 expression [104]. It has also been proven that ginseng increases the activity of NK cells [105]. Ginseng has been shown to protect mice against sepsis caused by *Staphylococcus aureus* by decreasing the secretion of inflammatory cytokines TNF-α, IL-1β, IL-6, IFN-γ, IL-12, and IL-18 [106]. Chan et al. (2011) focused on bird flu H9N2. They showed that ginsenosides (protopanaxatriol and ginsenoside Re) have protective properties against the H9N2 virus. Protopanaxatriol reduced the expression of IP10 (interferon gamma-induced protein 10), while the second ginsenoside reduces the DNA damage caused by the virus (H9N2-induced inflammation and apoptosis) [107].

### 4.3. Effects on Mothers

The effects of substances derived from ginseng on pregnant women are not yet fully understood. Xi et al. (2017) investigated the effect of supplementation of polysaccharides derived from ginseng root in a pregnant sow. The extract was given to sows from the 90th day of pregnancy to the 28th day after the birth of the progeny. There was no effect of ginseng supplementation on the total number of piglets, live piglets, weak piglets, and birth weight of piglets, which means that the herb had not affected the reproduction process. Moreover, ginseng treatment in pregnancy and lactation caused a significant increase in the total immunoglobulin G concentration in milk and serum of sows, which was associated with elevated levels of cytokines: IL-2, IL-6, TNF-α, and IFN-γ [44]. Taken together, the results present evidence that ginseng supplemented in pregnancy and lactation modulates the adaptive immunity of mothers.

Also, the positive effect of ginseng supplementation on the function immune system was reported by Concha et al. (1996). Authors investigated ginseng immunomodulatory effects isolated from peripheral blood or milk lymphocyte in an in vitro study. They showed that ginseng has a stimulating effect on both groups of isolated lymphocytes after pokeweed mitogen stimulation [45]. The same scholars have shown, a few years later, that subcutaneous injections of extract from the root of panax ginseng CA Meyer at a dose of 8 mg/kg body weight per day for 6 days can stimulate the innate immunity in cows with staphylococcus aureus infection. In ginseng-treated groups, the numbers of S. aureus-infected quarters and milk somatic cell counts tended to decrease. Moreover, the phagocytosis and oxidative burst activity of blood neutrophils isolated from cows treated with ginseng were significantly increased one week after injection, as well as the number of monocytes in blood [46]. These findings suggested that not only adaptive, but also innate immunity, may be affected after ginseng treatment in pregnant and lactating mothers.

### 4.4. Effects on Offspring

The supplementation of ginseng extract in the sow’s diet may have a beneficial effect on the development of immunity in newborns [44]. This is not only associated with passive immunity delivered to infants by milk (the higher ability of the mother’s immune system for the elimination of pathogens) but also with modulation in an immune response in newborns. Maternal supplementation of ginseng significantly increased IL-2 (an important part of organism response to microbial infection) and TNF-α (phagocytosis stimulant) concentration in the piglets’ serum.

Ginseng supplementation in pregnancy or the lactation period also plays a protective role against some negative factors. Administration of ginseng extract during pregnancy exposed to prenatal stress reduces the incidence of schizophrenia in the offspring [47]. According to the studies of Saadeldin et al. (2018), administration of an aqueous ginseng extract to pregnant females during exposure to phthalates and bisphenol A alleviates the toxic effects of these compounds in offspring [48]. Wanderly et al. (2013) studied if giving pregnant rats a ginseng extract could reverse the increase in testosterone production induced by dexamethasone. The extract was administered to females from 10 to 20 days of pregnancy by gavage, while dexamethasone was administered by injections from the 14th day to the 21st day of pregnancy. Adult male offspring were sacrificed, and blood, testis, and prostate were removed for further ex vivo examinations and morphological analysis. Plasma was obtained from the blood and the testosterone concentration was tested. The study showed that the ginseng extract is able to reverse the negative effect of dexamethasone on the synthesis of testosterone in Leydig cells [49].

The use of ginseng in pregnancy, however, has some limitations. The study by Belanger et al. (2016) showed that ginseng extracts have a negative effect on pregnancy in mice. Also, in vitro direct exposure to the ginseng extract reduced development in a concentration responsive manner [50].

## 5. Camellia

### 5.1. Characteristic and Immunomodulatory Properties

*Camellia sinensis* (black tea) [108] and *Camellia sinensis*, *Theaceae* (green tea) [109] are two of the most popular botanical plants. They are cultivated in over 30 countries, especially in tropical areas [53]. Infusions from tea are consumed in different countries, but its consumption is most popular in East Asia [110]. The main active ingredients of tea are polyphenols, more specifically catechins: epigallocatechin, epicatechin, and epicatechin gallate [111,112,113], which are characterized by antioxidant properties [112,114]. The most biologically active catechins are epigalocatechingallate [108].

Green tea can modulate macrophages and dendritic cells [115,116], as well as immunomodulatory properties of the immune system [117]. Tea can stimulate immunity by stimulating the secretion of antibodies [118,119]. It also acts as an antioxidant and has anticancer activity [120]. Black tea has a positive effect on the cardiovascular system [121]. Due to the limited data regarding the impact of tea consumption on pregnancy, caution is recommended [122]. High concentrations of black tea extract may be toxic during pregnancy [123].

### 5.2. Effects on Mothers

Losinskas-Hachul et al. (2018) [51] investigated the effect of green tea intake (400 mg/kg of body weight/day) by rat-mothers from the first day of pregnancy until the end of lactation on maternal and offspring metabolism. They showed that the intake of the extract by the mother increases the ratio of IL-10/TNF-α and IL-1β in mesenteric adipose tissue and causes a decrease in catalase activity in the liver. It has been proven that the intake of green tea extract during the lactation period inhibits the penetration of macrophages and increases the expression of AMPK (5’AMP-activated protein kinase), which affects the secretion of insulin [52]. A high dose of black tea extract (100 mg/kg of body weight/day) may contribute to changes in blood and kidneys of pregnant rats. The study by Dey et al. (2017) showed significant alterations in urinary calcium, creatinine, and urea during the prenatal period, while exhibited proteinuria, ketonuria, and histology showed nephrotoxicity during the postnatal period. The herb also affected concentrations of proinflammatory cytokines and decreased anti-inflammatory cytokines compared to the control group [53]. Moreover, the supplementation of black tea extract causes a decrease of hemoglobin concentration and loss of the biconcave structure of erythrocytes. It was also shown that the extract from black tea increases WBC levels in the mother’s blood and induced significant changes in the histology of liver and serum enzymes [54].

Kayiran et al. (2013) examined whether black tea consumed during pregnancy affects the oxidative/antioxidant status of breast milk. The mother’s milk was analyzed for lipid peroxidation based on malondialdehyde (MDA) levels and reduced glutathione levels (GSH). The study did not show a correlation between the amount of tea consumed and the level of MDA and GSH, which suggests that breast milk is insensitive to the antioxidants present in tea [124].

Deficiency of folic acid increases the risk of neural tube defects in the prenatal period. It has been shown that the levels of folic acid in serum is much lower in pregnant women who consumed more caffeine and tannins (in the form of coffee and oolong tea). According to Otake et al. (2018), pregnant women should minimize the consumption of drinks containing caffeine [55].

Epigallocatechin gallate (EGCG), a natural compound of green tea, has been shown to increase the efficacy of oral nifedipine treatment in severe pregnancy-induced preeclampsia [56]. On the other hand, excessive drinking tea during pregnancy may be associated with an increased risk of pre-eclampsia. Tea components affect this risk through a number of likely mechanisms, for example, as pathways associated with the modulation of angiogenic factors or with oxidative stress [57].

In the Jochum et al. (2017) study, pregnant women were given 300 ml of black tea and the content of flavonoids in milk was evaluated. It was shown that flavonoids (catechin, epicatechin) cannot be detected in milk samples, and the consumption of tea alone did not affect the total antioxidant capacity of breast milk [125].

### 5.3. Effects on Offspring

It has been shown that the consumption of green tea extract in pregnancy affects offspring’s health. The consumption of Japanese and Chinese tea during pregnancy is also associated with an increased risk of premature birth [58]. Yang et al. (2018) showed that drinking tea during pregnancy is a risk factor for low birth weight of offspring [59]. Camelia supplementation of mothers in the pregnancy or lactation period exhibits also health benefits. According to the research by Hachul et al. (2018), the mother’s intake of green tea extract exerts a protective action on the progeny that is on a high-fat diet. The extract protects against dyslipidemia, glucose intolerance, and fat accumulation. However, the mother’s intake of this extract has a pro-inflammatory effect on the adipose tissue of progeny that is not on a high-fat diet [60]. Also, Losinskas-Hachul et al. (2018) [51] observed this relationship in their rat study. The pump that fed rats the green tea extract was shown to decrease the retroperitoneal adipose tissue relative weight and SOD activity, but increased adiponectin, LPS, IL-10, IL-6 contentl and IL-10/TNF-α ratio in retroperitoneal, IL-10, and TNF-α content in gonadal, and IL-6 content in mesenteric adipose tissues. These changes indicated that the consumption of green tea extracts altered the inflammatory status of 28 day old offspring. Moreover, epigallocatechin 3-gallate supplementation improved the results of treatment of maternal and maternal gestational diabetes. It affects the reduction of neonatal complications, i.e., low birth weight and hypoglycemia, and alleviates the mother’s diabetic symptoms [61].

Tea extract showed also a protective role after exposure of compounds which have a negative influence on health. It has been showed that a moderate dose of green tea extract can decrease the number of malformations in fetuses after exposure to a teratogen, such as cyclophosphamide. However, too high a dose increases the toxicity of cyclophosphamide [62].

Due to the limited amount of research on the consumption of tea by pregnant and lactating women and the popularity of tea, further research is necessary.

## 6. Limitations in Medicinal Herbs Usage

There is no denying the fact that the use of medicinal herbs has its limitations and risks. Plant extracts may contain hundreds of compounds that have specific pharmacologic effects that may be synergistic or antagonistic [126]. Ginseng extract showed contradictory effects in animals, such as histamine and antihistamine-like actions, hypertensive and hypotensive effects, and stimulatory or depressant activity on the central nervous system [127,128]. The synergistic effects of the plant compounds have been shown in Cinchona, which has almost 30 alkaloids [129]. Four of those, cinchonine, l-isomer cinchonidine, quinine, and d-isomer quinidine, have antiplasmodial activity [130]. However, Druilhe et al. (1988) showed that the mixture of alkaloids is two to ten times more effective in vitro than any of the alkaloids used separately [131]. Differences in the bioactivity of extracts of the same plant may result from differences in the composition and concentration of active compounds. However, studies investigating substance that differentiates the extracts often do not lead to the discovery of the active substance responsible for the biological effect, such as in the case of Rhodiola and epigallocatechin [132]. Therefore, it is possible that the positive effects of medicinal herbs are based on the complex relationship between pharmacologic effects of the herb’s compounds, and usage of the single compound would not have any significant therapeutic effects.

Also, it is unpractical to isolate and assess every active ingredient from the herbal extract, since the resources required would be huge [133]. Isolation and investigation of every active compound in a plant extract would be laborious and might not render any meaningful results due to synergistic effects between plant compounds.

As another limitation of medicinal herbs usage, one has to consider geographical differences and potential contamination of the extract resulting from the culture, collection, and storage of the herbs [134]. Variation in the secondary metabolites content between plants may be caused by the region of origin, season, cultivar, and nitrogen availability [135,136]. For example, Marrassini et al. (2018) showed that total polyphenols, flavonoids, and tannins content in Urera aurantiaca obtained from two different regions of the same country is significantly different [137].

Therefore, since a simple medication must be a single substance with a proven therapeutic effect, it may never be possible to register any medical herb as a drug, and another unified approach must be implemented for the registration and pharmacovigilance of medicinal herbs.

## 7. Conclusions

The results presented here suggest that some of the immunomodulators, supplemented in pregnancy and lactation, may affect offspring health. Therefore, before the recommendation to use plant supplementation in the maternal diet to enhance immune system function, the safety of its use should be determined in tests performed on in vivo models. Ideally, if the mechanism of action of these compounds was determined, or the isolation of the substance responsible for the positive biological effect of the plant was achieved. Unfortunately, there is still insufficient knowledge in this topic, which can explain the lack of studies on the effects of medicinal herbs supplementation on the health of pregnant women and their offspring in the database of the national institutes of health (www.clinicaltrials.gov).

## Figures and Tables

**Table 1 nutrients-11-01958-t001:** Summarized effects of *Rhodiola*, *Echinacea*, Ginseng, and Camellia, or its extracts, supplemented in pregnancy and/or lactation on mothers and offspring health.

Herb or Extract	Key Substances	Pharmacological Action	
		Mother	Offspring
*Rhodiola*	phenylethanoid salidroside and tyrosol, phenolic acids (i.e., chlorogenic, ferulic, ellagic and p-coumaric), and flavonoids (i.e., fisetin, naringenin, kaempferol, epicatechin, luteolin, quercetin, epigallocatechin and (+)-catechin)	reduces the percentage of cells with a respiratory burst in granulocytes (supplementation with RKW) [28] increases in the percentage of granulocytes and monocytes in the blood with the respiratory burst (supplementation with RKW-A) [28] contributes to changes in spleen morphology and structure [29] increases the concentration of VEGF and bFGF [30] reduces the number of CD4 + and CD19 + cells and the total number of NK cells [31]	increases hemoglobin concentration (about 0.6 mg/dL) [32] decreases in the mean percentage of lymphocytes in peripheral blood, and an increase in the mean percentage of granulocytes [32] decreases in the percentage of CD3+ cells and CD4+ [32] increases the concentration of IL-10 in the serum [33] stimulate the phagocytosis process [32] significant difference in tissue localization and the number of CD8+ cells [34] contributes to a higher number of CD8+ cells in the central part of the spleen [34] influence cell proliferation in response to mitogen supplementation (LPS, PHA and ConA) [29] decreases the number of apoptotic cells [35] decreases the concentration of VEGF in the sera [30]
*Echinacea*	alkamides, ketoalkenes, caffeic acid derivatives, polysaccharides, glycoproteins, and caftaric acid	does not affect hematological and reproductive parameters [36] no influences on the enzyme results [37] decreases the level of crude protein in colostrum [37] decreases the level of antibodies in the plasma [37] decreases the number of spleen lymphocytes and nucleated erythroid cells [38] contributes to more frequent miscarriages in the early stages of pregnancy [38]	decreases the number of embryos in litter and significantly diminished VEGF and bFGF content of embryos tissue [39] increases phagocytic activity in blood [40] increases bacterial diversity [40] non-teratogenic, does not increase the risk of malformations [41,42,43]
Ginseng	polysaccharides, flavonoids, fatty acids, peptides, and saponins (mainly ginsenosides)	increases the total IgG concentration in milk and serum of sows, which was associated with elevated levels of cytokines: IL-2, IL-6, TNF- α, and IFN-γ [44] stimulates the effect of isolated lymphocytes after pokeweed mitogen stimulation [45] stimulates the innate immunity in cows with *Staphylococcus aureus* infection [46] increases phagocytosis, oxidative burst activity of blood neutrophils and number of monocytes [46]	increases IL-2 and TNF-α concentration in the piglets’ serum [44] reduces the incidence of schizophrenia in the offspring [47] alleviates the toxic effects of phthalates and bisphenol A [48] reverses the negative effect of dexamethasone on the synthesis of testosterone in Leydig cells [49] teratogenic effect [50]
Camellia	epigallocatechin, epicatechin, epicatechin gallate	increases the ratio of IL-10/TNF-α and IL-1β in mesenteric adipose tissue and causes a decrease in catalase in the liver [51] inhibits the penetration of macrophages and increases the expression of AMPK (during lactation) [52] contributes to alterations in urinary calcium, creatinine, and urea during the prenatal period, nephrotoxicity [53] increases levels of proinflammatory cytokines and decreases anti-inflammatory cytokines levels in serum [53] decreases of hemoglobin concentration and loss of the biconcave structure of erythrocytes [54] increases of WBC level in the mother’s blood and induced significant changes in the histology of liver and serum enzymes [54] decreases the level of folic acid [55] increases the efficacy of oral nifedipine treatment in severe pregnancy-induced preeclampsia [56] may be associated with an increased risk of pre-eclampsia [57]	increases the risk of premature birth [58] risk factor for low birth weight of offspring [59] protect against dyslipidemia, glucose intolerance, and fat accumulation [60] pro-inflammatory effect on the adipose tissue (not on a high-fat diet) [60] decreases the retroperitoneal adipose tissue relative weight and SOD activity but increases adiponectin, LPS, IL-10 and IL-6 content and IL-10/TNF-α ratio in retroperitoneal, IL-10 and TNF-α content in gonadal, and IL-6 content in mesenteric adipose tissues [51] improves the results of treatment of maternal gestational diabetes [61] reduces neonatal complications [61] can decreases the number of malformations in fetuses after exposure to cyclophosphamide, but too high dose increases the toxicity of cyclophosphamide [62]

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
