# Peer review of "Supplementation of Plants with Immunomodulatory Properties during Pregnancy and Lactation—Maternal and Offspring Health Effects"

_nutrients, 2019, doi:10.3390/nu11081958_

Round 1

Reviewer 1 Report

The manuscript “Supplementation of plant immunomodulatory in pregnancy and lactation – effect on mothers and offspring health” is an interesting bibliographic review, but requires  some considerations.

Currently there is great interest in the scientific community and consumers in general on the treatment with medicinal herbs, which also extends to pregnancy and lactation. However, there is no proven security on its use.

The authors present a review of data from animal experiments on the four most studied immunomodulatory plants that have been used in maternal nutrition.

The authors have previously studied the subject and provide many bibliographic citations of their own.

Please find below a list of more specific issues that need to be addressed throughout the manuscript:

The manuscript would have better visualization if tables of summary of the evaluated findings were provided.

The authors should discuss more about some concepts.

- A simple medication must be a single drug substance that has a therapeutic effect, however, in a plant there may be several substances (with additive and opposite effects).

- Likewise, it should be considered that the effect of therapy with medicinal plants may also be affected because in the culture, collection or storage may be contaminated by other plants.

- The final form of preparation of medicinal plants for consumption can vary their final pharmacological characteristics. This is very important, taking into account that traditional use may be different in each geographical area.

- The final form of preparation of medicinal plants for consumption can vary their final pharmacological characteristics. This is very important, taking into account that traditional use may be different in each geographical area.

On line 421, www.clinicalrials.gov should be changed to www.clinicaltrials.gov

Authors should consult the Nutrients journal standards for the use of references in the text and in the references section.

Author Response

Response to the Reviews

All changes in the manuscript text according to all reviewer suggestions are marked in red.

Review 1:

The manuscript would have better visualization if tables of summary of the evaluated findings were provided.

Response: We added the table 1. The effect of herb or extract on the mother and offspring health.

The authors should discuss more about some concepts.

- A simple medication must be a single drug substance that has a therapeutic effect, however, in a plant there may be several substances (with additive and opposite effects).

- Likewise, it should be considered that the effect of therapy with medicinal plants may also be affected because in the culture, collection or storage may be contaminated by other plants.

- The final form of preparation of medicinal plants for consumption can vary their final pharmacological characteristics. This is very important, taking into account that traditional use may be different in each geographical area.

Response: We added required information (with references) to the new paragraph: “limitations of medical herbs usage”.

On line 421, www.clinicalrials.gov should be changed to www.clinicaltrials.gov

Response: We changed it.

Authors should consult the Nutrients journal standards for the use of references in the text and in the references section.

Response:  References in paper were changed with accordance to the Journal requirements.

Reviewer 2 Report

This is a very interesting review on the effects of 4 common plants with immunomodulatory roles on maternal and offspring health. 

The authors provide a rich review of the literature with interesting results. 

Main issue: 

The main issue is that the results should be better presented. Indeed, I suggest that the authors should use the same sub-titles for each « plant » paragraph (such as the ones used for Rhodiola). As a result, the construction of the paragraphs (for Echnacea, Ginseng and Camellia) should be re-organized. 

Minor issues: 

Review the title (for example: « Supplementation of plants with immumomudulatory properties during pregnancy and lacatation - maternal and offspring health effects ») P3, L135 : please correct « are necessary » (instead of « is necessary »). P5 L136 : delete the dot in the title P5 L240-245 : abbreviation explanations should be mentionned ealier : since  P2, L89-90. Abbreviation explanation is missing p7 L315 (tlr4) and L317 (tlr2)

Author Response

Review 2:

Main issue: The main issue is that the results should be better presented. Indeed, I suggest that the authors should use the same sub-titles for each « plant » paragraph (such as the ones used for Rhodiola). As a result, the construction of the paragraphs (for Echnacea, Ginseng and Camellia) should be re-organized.

Response: We rewrote the paragraphs for Echinacea, Ginseng and Camellia with accordance to the suggestion.  

Minor issues: 

Review the title (for example: « Supplementation of plants with immunomodulatory properties during pregnancy and lactation - maternal and offspring health effects »)

Response: We changed the title.

P3, L135 : please correct « are necessary » (instead of « is necessary »). P5 L136 : delete the dot in the title

Response: We corrected these sentences.

P5 L240-245 : abbreviation explanations should be mentionned ealier : since P2, L89-90.Abbreviation explanation is missing p7 L315 (tlr4) and L317 (tlr2)

 Academic reviewer

The review provides potentially interesting information, but would be strenghtened by the inclusion of evidence in humans, which exists for some of the compounds examined or their sources (e.g. tea).

Response: We added required information to the Tea paragraph, and others.

 Round 2

Reviewer 2 Report

Dear authors,

Thank you very much for correcting your very interesting paper.

Best regards